# Bio-Artificial Liver Support System: A Prospective Future Therapy

Chyntia Olivia Maurine Jasirwan [1,*,†] , Akhmadu Muradi [2,†] and Radiana Dhewayani Antarianto [3,†]

1   Hepatobiliary Division, Medical Staff Group of Internal Medicine, Faculty of Medicine, Universitas Indonesia, Dr. Cipto Mangunkusumo Hospital, Jakarta 10430, Indonesia
2   Vascular and Endovascular Division, Medical Staff Group of Surgery, Dr. Cipto Mangunkusumo Hospital, Jakarta 10430, Indonesia
3   Department of Histology, Faculty of Medicine, Universitas Indonesia, Jakarta 10430, Indonesia
*   Correspondence: chynmadu@gmail.com or hepatologibilierfkuirscm@gmail.com
†   These authors contributed equally to this work.

**Abstract:** Whether acute or chronic, liver failure is a state of liver dysfunction that can progress to multiorgan failure. Mortality in liver failure patients is approximately 80–90% and is caused by detoxification failure, which triggers other immediate complications, such as encephalopathy, coagulopathy, jaundice, cholestasis, and acute kidney failure. The ideal treatment for liver failure is liver transplantation, but the long waiting period for the right donor match causes unavoidable deaths in most patients. Therefore, new therapies, such as tissue engineering, hepatocyte transplantation, and stem cells, are now being studied to anticipate the patient's condition while waiting for liver transplantation. This literature review investigated the effectiveness of some bio-artificial liver support systems using review methods systematically from international publication sites, including PubMed, using keywords, such as bio-artificial liver, acute and chronic liver failure, extracorporeal liver support system (ECLS), MARS, single-pass albumin dialysis (SPAD). Artificial and bioartificial liver systems can show specific detoxification abilities and pathophysiological improvements in liver failure patients but cannot reach the ideal criteria for actual liver function. The liver support system must provide the metabolic and synthetic function as in the actual liver while reducing the pathophysiological changes in liver failure. Aspects of safety, cost efficiency, and practicality are also considered. Identifying the technology to produce high-quality hepatocytes on a big scale is essential as a medium to replace failing liver cells. An increase in detoxification capacity and therapeutic effectiveness must also focus on patient survival and the ability to perform liver transplantation.

**Keywords:** bio-artificial liver support system; ECLS; MARS; liver failure; SPAD

## 1. Introduction

Liver failure (LF) is liver dysfunction defined by coagulopathy, jaundice, and liver encephalopathy [1]. A patient with LF has high mortality ranging from 80–90%, some with fulminant liver failure [2]. The loss of liver function, including detoxification, biotransformation, excretion, and secretion, progresses to severe clinical syndrome causing organ failure and death [3]. Liver failure is usually caused by toxins, such as acetaminophen, ischemia, or infection. Other common liver failure complications are encephalopathy, coagulopathy, jaundice, cholestasis, pruritus, sepsis, and acute kidney failure [4]. Although some patients may spontaneously recover, many will die while waiting for a suitable donor due to damage to liver function. The widening gap between the number of patients on the waiting list with the number of liver donors has highlighted the vital need for alternative therapies for LF.

There is much enthusiasm for developing artificial and bioartificial support systems, which have gone through many clinical trials [5]. The artificial ideal support system must detoxify wastes, such as ammonia, provide the albumin synthetic function and coagulation factors, reduce inflammation, and promote cell regeneration. Cell transplantation, including

hepatocytes, stem cells, and liver from tissue engineering, is currently being studied [6]. There are two kinds of artificial/extracorporeal liver support (ECLS) therapies that can be further divided into artificial or bioartificial (bioreactor) kinds, with the latter mainly focusing on metabolic stabilization and detoxification. The ECLS system is based on a dialysis technique to remove hepatotoxic substances such as cytokine, vasoactive materials, endotoxins from gut flora, nitric oxide, prostaglandin, reactive oxygen species, toxins with low molecular weight, and other pathogen-related molecules, which caused the liver failure pathogenesis. They used the free cell technique for plasma filtration [7]. The ECLS equipment combined a bioreactor for cell perfusion with the patient's plasma, consisting of a three-dimensional matrix intertwined with porous fiber embedded with human hepatocytes [8], a hemodiafiltration, and SPAD [9]. Clinical studies with ECLS were still limited to small trial studies. No safety issues were recorded in 11 patients treated with ECLS, such as hemodynamic instability, complement activity, or organ function decline [10]. In a controlled but not random trial in 17 patients with acute liver failure, ECLS therapy was correlated to encephalopathy improvement without harming hemodynamic parameters. However, ECLS was discontinued in two patients with worsening intravascular coagulation [11]. Although it can be used as a bridge for transplantation, it is unclear if ECLS can improve survival for patients who are not transplantation candidates [12]. ECLS can be divided into artificial and bio-artificial systems.

Living cells, sourced from pigs or humans, are loaded into extracorporeal bioreactors in bioartificial liver support systems. No bioartificial system is yet commercially accessible outside of clinical studies. Blood purification is the main idea behind various artificial liver support options. They included extracorporeal liver support (ECLS), molecular adsorbent recirculation system (MARS); fractionated plasma separation and adsorption (SPAD); bio-artificial liver system (BALS); induced pluripotent stem cells (iPSCs) [13]. They have been developed to complete the liver function, bridge the patient toward liver transplantation, or enable the original liver to regenerate [2].

This study aimed to conduct a literature review investigating the difference between the types of liver support systems, such as single-pass albumin dialysis (SPAD), extracorporeal liver support (ECLS), molecular adsorbent recirculating system (MARS), bio-artificial liver system (BALS), induced pluripotent stem cells (iPSCs) in liver failure both in human or animal studies with sources from international publication sites, such as PubMed. The keywords used were bio-artificial liver, acute liver failure, chronic liver failure, extracorporeal liver support (ECLS), molecular adsorbent recirculation system (MARS), single-pass albumin dialysis (SPAD), induced pluripotent stem cells. The studies found 37 articles, eight systematic reviews, and 29 clinical research articles.

## 2. Materials and Methods

Through the literature review, a significant amount of research in numerous fields has been discovered and validated as a method for examining and analyzing problems objectively. Therefore, the literature review aims to gather, interpret, evaluate carefully, and identify all current studies pertinent to a predetermined scope of investigations to give research groups extensive knowledge.

The research questions (RQ) addressed through this literature review are given below:

1. What is the method of liver support system for treating chronic and acute liver failure?
2. How do the advantages and disadvantages of each conventional liver support system provide liver failure treatment?
3. What is the bio-artificial liver support system method, and how can it treat liver failure compared to the other systems?

For a methodical writing survey, arranging and directing a formal pursuit process is vital. A sorted-out pursuit process makes it conceivable to exhume all the accessible advanced assets keeping in mind the goal of locating all related accessible writing that meet the required criteria. For this study, an inquiry has been led to discovering important papers in meeting procedures, books, journals, conferences, and other online materials. In

the present study, several keywords related to the design and estimation of waste plastics based on the research questions (provided in the Research questions section) were searched in the digital libraries mentioned below:

a. ScienceDirect (http://www.sciencedirect.com) (accessed on 15 June 2022)
b. SpringerLink (http://www.springer.com/in/) (accessed on 25 June 2022)
c. ProQuest (http://www.proquest.com) (accessed on 30 June 2022)
d. PubMed (https://pubmed.ncbi.nlm.nih.gov/) (accessed on 25 June 2022)
e. Elsevier (https://www.elsevier.com/en-xs) (accessed on 10 June 2022)
f. Cochrane (https://www.cochrane.org/) (accessed on 15 June 2022)

The keywords used are bio-artificial liver, acute liver failure, chronic liver failure, extracorporeal liver support (ECLS), molecular adsorbent recirculation system (MARS), single-pass albumin dialysis (SPAD), induced pluripotent stem cells. Using merely the keyword "liver support system," searches turned up most of the studies. Additional keyword strings built using the phrases "OR" and "AND" were also used to ensure no pertinent publication was overlooked. The suggested study period covered the latest ten years (2012–2022). The search turned up a significant amount of the literature in the form of journal articles, conference proceedings, and other published works, including books and periodicals. Using predetermined keywords, each of the included digital repositories was manually searched. Using Mendeley software, the relevant citations and bibliographic data were considered carefully. In the first search, it was chosen to keep a different library for each digital source, and after filtering and excluding duplicates, all the libraries were combined into a single file library. This bibliographic information includes the name(s) of the author(s), the article's title, the name of the journal or conference, the year the article was published, and the number of pages in the paper.

After filtering, a list containing a total of 202 references was managed in the file of the Mendeley library. The details of the overall search process in the specified digital libraries are outlined in Figure 1. A total of 350 titles were found. First, the duplications in these publications (more than one version of the paper) were removed. After that, the papers were checked manually and filtered by titles, abstracts, and, finally, the contents. The final filter results reached 37 articles, of which 8 were systematic review articles and 29 were clinical research articles.

Research Questions and Searching Process
(total 83 articles, 25 literature review and 58 original research articles)

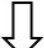

Study Selection based on Inclusion and Exclusion Criteria (total 37 articles: 8 literature review and 29 original research articles) through screening the article starts from the titles, abstract, methods and conclusion part.

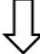

Data Extraction and Analysis

**Figure 1.** Protocol study for the literature review.

## 3. Results

### 3.1. Liver Support System

#### 3.1.1. Molecular Adsorbent Recirculation System (MARS)

A molecular adsorbent recirculation system (MARS) is a liver support system involving two circuits: one with a high-flux dialyzer impermeable to albumin and a secondary system filled with albumin solution [14]. This system is designed as a multisystem purification with continuous albumin-bound toxin detoxification based on dialysis, filtration, and adsorption principles. The blood initially passes through a closed circulation separated from the patient's blood by the high-flux bio-compatible filter coated with albumin by removing water soluble and albumin-bound toxins [15]. The albumin is dialyzed using 5–10% standard dialysis solution in continuous venous hemodialysis over an albumin impermeable membrane. Exogenous albumin dialysate is then regenerated using dialysis with conventional dialysate solution, and further absorption is at the anion and charcoal exchange resin columns. Regenerated albumin dialysate is recirculated to repeat the process. The albumin-enriched dialysate (20% albumin) circulates counter-wise to the blood to transfer albumin-bound toxins to the dialysate. Albumin-bound toxins in the plasma were forwarded to the membrane, which will be absorbed and taken by the dialysate. Substances with molecular weight exceeding 50 kDa are not removed due to the size of the membrane pores [16]. The length of time for regular therapy is 6–8 h per day [17].

Many previous studies have shown the ability of MARS to eliminate toxins, repair liver failure symptoms, and increase the survival rate during acute liver failure (ALF). Clinical usage of MARS has consistently been correlated with a reduction in bilirubin, bile acid, ammonia, urea, lactate, and creatinine levels in various patient studies with acute and chronic liver failure. The benefits of MARS were investigated in another random trial in 23 patients with acute chronic liver failure with progressive hyperbilirubinemia. The bile acid and plasma bilirubin levels decreased significantly in the MARS group [18].

MARS therapy can stabilize the anhepatic condition in ALF and ACLF for 96 h and later bridge the patient for liver transplantation. A case study by Dacha et al. in 2014 mentioned a 56-year-old female ALF patient who started MARS therapy with hepatic encephalopathy class 4, severe coagulopathy, severe metabolic acidosis, and 90% liver necrosis had experienced stability in cardiopulmonary, neurological, and metabolic after 96 h of MARS therapy so she could successfully undergo liver transplantation. The second case was a 67-year-old male patient with severe ACLF by profound coagulopathy and grade 4 hepatic encephalopathy who maintained clinical stabilization and successfully received liver transplantation after 96 h of MARS treatment [19].

A study in vitro that used 1 L of human plasma showed that MARS was more effective in clearing bile acid than SPAD (albumin concentration of 4.5%) at a similar dialysate flow rate [20]. MARS treatment was correlated with a significant increase in mean arterial pressure and systemic blood circulation resistance without a change in heart index [21]. The favorable hemodynamic response by MARS was correlated with plasma renin activity reduction and aldosterone, norepinephrine, and vasopressin levels [9].

However, the survival level for MARS has not been reported in prospective random clinical trials [21]. A MARS-related random study in patients with acute chronic liver failure mentioned that MARS was insignificant in reducing the 28-day survival rate after adjusting for age, systolic blood pressure, or the end-stage liver disease model (MELD) score. Although the MARS group's bilirubin and creatinine levels increased, the mortality (40.8% vs. 40%) did not differ for 30 days [22].

However, MARS's lack of metabolic and synthetic activity limits effectivity and can increase thrombocytopenia, coagulopathy, and bleeding risks. In a retrospective review of 83 MARS therapies in 21 patients, nine coagulopathy, bleeding, or thrombocytopenia episodes occurred [23]. However, still, MARS has a good safety profile. A significant challenge for MARS is that it is relatively more expensive per patient than other treatments [14].

### 3.1.2. Single-Pass Albumin Dialysis (SPAD)

Single-pass albumin dialysis (SPAD) is a form of simplified albumin dialysis using the available continuous venovenous hemodialysis equipment and only adding the albumin into the conventional dialysis solution [24]. Albumin dialysate is not regenerated or recycled, which coined the term "single-pass". The albumin concentration in this technique is significantly lower compared to MARS (2–5% albumin). The dialysate fluid flow is usually at 20–30 mL/minute with a tending time of 6–10 h per day. The SPAD system is simple and available. A significant limitation is a need for extra expenses to add albumin to the dialysate [9].

In the previous study, using SPAD as a 2% or 5% albumin solution for 6 h significantly reduced the total bilirubin level compared to conventional hemodialysis. Twelve consecutive patients with acute or chronic liver failure and hyperbilirubinemia (total bilirubin > 20 mg/dL) were treated with SPAD 2% for 6 h per day. A significant reduction was observed in total plasma bilirubin, conjugated bilirubin, blood urea nitrogen, and creatinine [9]. However, the clinical experience, including the effectivity of toxin removal with SPAD in treating liver failure, is still limited compared to the MARS system [9].

### 3.2. Bio-Artificial Liver Support System

Bio-artificial liver support system (BALSS) can improve ALF patients' survival by providing partial liver function until the matching liver donor is found or the original liver regenerates. Some types of BALSS have been applied to treat ALF patients in phase I studies or controlled clinical trials and showed improvement of the neurological state and liver and kidney functions to bridge transplantation or aid spontaneous recovery [25].

BALSS aims to combine blood purification with synthetic liver function replacement using a bio-reactor that contains hepatic cells into pure mechanical artificial and albumin-based dialysis. BALSS depends on a hemofilter (membrane cut off at 100 kDa) with hepatocytes embedded in the extra capillary compartment of the porous fiber cartridge. All blood, not plasma, passes through the fibers after re-heating and oxygenation. The porous material must enable the passing of small substances, such as toxins and synthesized proteins, and must separate the cells from the blood or plasma to avoid immune rejection. Most BAL systems consisted of porous fiber cartridges packed with single hepatocytes. The system is often combined with a detoxification step (charcoal or anion exchange resin) and an oxygenator [26]. Bio-artificial liver (BAL) consists of live cells that perform normal liver functions. They must be isolated and immobilized at suitable platforms, which maintain the cells' morphology and metabolism. Oxygen and nutrition must be available at the right concentration. Cells usually originate from hepatoblastoma cell lines or the pig liver. Ideally, the BAL support system will use primary human cells, but many high-quality human hepatocytes are hard to find. Human cells also have tumorigenic potentials, are slow growing, lose P450 function over time, and are challenging to distribute widely to other treatment locations. By contrast, primary pig liver hepatocytes are the most natural cell source because of the availability advantage and easy distribution [22].

### 3.2.1. Animal Subject Study

A previous study from Chen Y (2012) uses 100 g primary pig hepatocytes in one porous fiber cartridge to bridge six out of seven patients with acute liver failure to liver transplantation, with one who recovered without transplantation [27]. However, the hepatocyte-based BAL system raises concerns for xenozoonotic and retroviral transmission, but there has been no zoonosis infection [28].

BAL has various types with different cell sources, cell masses, culture methods, and architectural designs, such as bio-reactors, scaffolds, and separators, which might be related to the BAL's effects and safety. Furthermore, BAL has been pre-clinically modified in large animals but has not yet been used in clinical trials, which caused a significant gap in clinical and preclinical studies [25].

### 3.2.2. Human Subject Study

A six patient randomized controlled trial (RCT) meta-analysis in previous report also concluded that the BALSS might not reduce ALF deaths [21]. The survival results and side effects of this alternative method are still controversial [29].

Besides providing clinical efficacy, BAL faces some challenges in regulation and technicality. The use of live cells is complicated further by the need to maintain metabolism and cell function, avoid immunity activation, and prevent zoonosis disease transmission. The delivery of a bioreactor tends to be expensive and impractical. BALSS and its potential side effects are limited in case reports or series [30]. There has not yet been a study to evaluate the death effect of BALSS. There is not yet an ideal artificial system for routine use in patients with liver failure [31].

Wu Guohua et al. (2020) developed a new bio-artificial liver device to address the challenges of less-optimum detoxification by hepatocytes. A schematic of the new BALSS with an entire bioengineered liver based on DLM/GelMA for potential HE prevention is shown in Figure 2 below. Hepatocytes in people with liver failure cannot biotransform ammonia from the intestine, and if too much ammonia reaches the brain through the blood-brain barrier, it can lead to HE. GelMA and HepG2 cells were added to a DLM and constantly cultivated in an oxygenated bioreactor to create a complete bioengineered liver. With vastly improved biosynthesis and biotransformation capacities, the DLM/GelMA-based cells bioengineered an entire liver and offer excellent potential to help patients with liver failure and stop the development of HE [32].

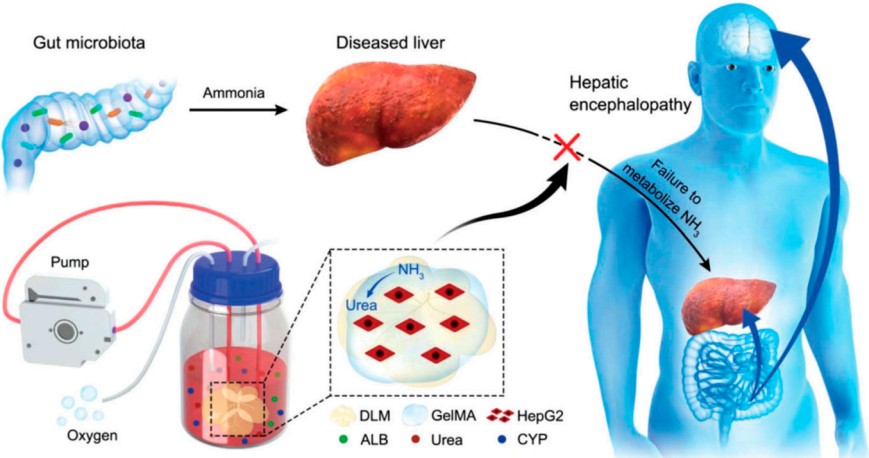

**Figure 2.** New bio-artificial liver devices development [32].

### 3.3. Induced Pluripotent Stem Cells/iPSC

Induced pluripotent stem cell-based tissue (iPSC) can be an alternative to primary hepatocyte cell therapy [33]. The paracrine effect of stem cells can modulate the local tissue and hasten the liver recovery process, while liver differentiation enables the preparation of regular liver reconstruction.

Liver failure therapy using stem cells might mainly depend on cell differentiation because the stem cell has genetic abnormalities. Co-culture using mesenchymal stem cells (MSC) can provide primary human hepatocytes support, increasing its viability and function, and becomes potential for the primary hepatocytes sources in bio-artificial liver devices [34].

Hepatocyte-like cells (HLCs) derived from stem cells with or without culture using mesenchymal tissue showed promising results [35]. The iPSC culture can enable a high-density, large cell production scale and increase the functional maturity and longevity of HLC [36].

Animal Subject Study

A previous study by Pourcher et al. [33] showed that tissue engineering using decellularized liver stem cells is a potential strategy for liver failure treatment because the liver's architecture is well maintained [37]. An in vivo study by Wang et al. [38] showed partial liver lobe decellularization stem cells in mice with a survival rate through SD 1% solution perfusion, representing a promising strategy for liver engineering and regeneration [39]. Therefore, recellularization using a stem cell aggregate can facilitate recovery of normal liver function and complex liver architecture. Li et al. [40] previously mentioned that the increase in liver differentiation and stem cell maintenance with similar morphology and viability to hepatocytes could use the human umbilical cord-derived mesenchymal stem/stromal cell (hUC-MSC) aggregate or mesenchymal/stromal tissue from a human umbilical cord, which was cultured in a decellularized pig liver scaffold for liver tissue engineering and regeneration [40].

Until now, no stem cell-based BAL system has been tested on humans. The method for functional hepatocyte production on a large scale must be developed in the future so the BAL system can enter the clinical phase. The continuous advance in iPSC hepatocytic differentiation might produce metabolically active functional HLC, which is needed for the BAL system in the future [41].

Generally, the liver cells from pluripotent stem cells are not yet matured, as noted in the cell's alpha-fetoprotein (AFP). In the adult liver, the AFP level is deficient. Studies have shown that stem cell-derived HLCs are phenotypically and functionally more similar to fetal hepatocytes than adult human cells [42]. Historically, the HLCs produce lower albumin, cytochrome P450, and urea cycle and have a consistently higher AFP level [43,44]. Although, the iPSC-derived HLC can still express specific hepatocyte markers, such as glycogen and lipid deposit activity, albumin secretion, and CYP450 metabolism activity, and improve the damaged mice's livers' functional states after transplantation [45]. However, the CYP metabolic process and activity produced by HLCs from the liver bio-artificial system are adequate for human in vitro models for toxicity testing and drug studies [46]. The drug toxicity test and metabolic profiling can use iPSC to predict the difference between individuals in liver metabolism and drug sensitivity mediated by genetic polymorphism, which affects drug efficacy and side effects [47], especially in liver injury induced by idiosyncratic drugs. Certain liver diseases caused by genetic disorders are manageable using the iPSCs model approach and possibly offer improved genetic correction [44].

The primary safety issue of HLC cells in tumorigenesis has mainly been managed with direct cell reprogramming. However, although they have been reprogramming by not involving genome integration, the change in the basal reprogramming factor expression has been reported to be cancer-related [46].

A summary of the advantages and disadvantages of the liver support system methods is shown in Tables 1 and 2 below.

Clinical trial design for BAL devices is a significant challenge. In particular, the course of liver failure is highly variable and depends on the etiology. In addition, hepatic encephalopathy, one of the main manifestations of liver failure, is challenging to quantify clinically. Therefore, patients should be randomized in clinical trials while controlling for the stage at which support was started and the individual etiology. Likewise, determining the appropriate controller therapy can be difficult.

Ideally, to minimize the non-specific effects of extracorporeal treatment, a non-biological control, such as venovenous dialysis, can be used. Another challenge is the choice of the clinical endpoint. Most clinical studies use the endpoints 30-day survival and 30-day transplant-free survival; however, studies can often be confounded by the fact that patients with acute liver failure receive different transplants at a given center depending on organ availability and established selection criteria

**Table 1.** Advantages and Disadvantages of Each Liver Support-System Method.

| No. | Liver Support Method | Advantages | Disadvantages |
|---|---|---|---|
| 1. | Single-Pass Albumin Dialysis (SPAD) | Significant reduction (>50%) in total plasma bilirubin, conjugated bilirubin, blood urea nitrogen, and creatinine | Clinical studies about effectiveness in toxin removal are still limited (only 1–2 studies from the last five years). |
| 3. | Molecular Adsorbent Recirculation System (MARS) | Eliminate toxins, and repair liver failure symptoms (toxin blood level 0%) Reduction in bilirubin, bile acid, ammonia, urea, lactate, and creatinine levels (>50% compared to previous levels) MARS can stabilize the anhepatic condition in ALF and ACLF for 96 h before transplantation. Effective in clearing bile acid compared to SPAD (bile acid clear 100%) A significant increase (50%) in mean arterial pressure and systemic blood circulation resistance without a change in heart index. Plasma renin activity reduction (up to 50%). | The survival rate has not been reported (no studies). The mortality percentage did not significantly reduce (compared to the previous study, 0% reduction). Thrombocytopenia, coagulopathy, and bleeding risks up to 20–30%. MARS is relatively more expensive (cost >1000 USD in Europe) |
| 4. | Bio-artificial Liver Support System (BALSS) | Improve neurological state and liver and kidney functions to bridge transplantation (the function status is usually based on assessment score). | BALSS might not reduce ALF deaths, but the survival rate is still controversial. It can either reduce or increase ALF deaths (the percentages are 50%) The delivery of bioreactors is expensive and impractical (it costs USD >500) Limited study and case reports (there have only been 1–2 studies in the last five years.) The risk of zoonosis disease transmission is up to 30%. |
| 5. | Induced-Pluripotent Stem Cells/iPSC | Increase the viability and function of human hepatocytes (from 20 to >40%) Improve the regeneration of liver function and facilitate a speedy recovery (up to 100%). | There is no study with human subjects reported. Therefore, the method should be produced on a larger scale of up to <100 subjects. |

**Table 2.** The Comparisons of Human and Animal Subjects for Liver Support Study [10–19].

| No. | Liver Support Method | Human Subject | Animal Subject |
|---|---|---|---|
| 1. | Hepatocyte-based Bio-Artificial Liver (BAL) | Not yet used in clinical trials. RCT meta-analysis results showed that BAL might not reduce ALF deaths. | Pre-clinically modified and applied to large animals. As a result, six out of seven patients with liver failure bridged to transplantations, and one recovered without transplantation. |
| 2. | Single-Pass Albumin Dialysis | Twelve patients with ALF and CLF were treated, and there was a reduction in total bilirubin levels. However, clinical experience in toxic removal is still limited. [48] | One of the simplest methods for eliminating toxins and water-soluble compounds from albumin has been proven to be single-pass albumin dialysis (SPAD), which was tested on five pigs. |
| 3. | Molecular Adsorbent Recirculation System (MARS) | A random clinical trial in 23 patients with ALF and CLF showed a reduction in bilirubin level. MARS has been proven to eliminate toxins and stabilize the anhepatic condition in humans. | |

The presence of non-biological add-ons, such as charcoal perfusion, are in some designs. A direct comparison of the effect of non-biological systems alone, dead or non-hepatocyte cells, and live hepatocytes would provide essential information on the effectiveness of cellular components, particularly since dead hepatocytes and non-hepatocyte cells have been shown to offer some survival benefits in various animal models of acute liver failure.

In particular, the ability to assess cell viability and function more accurately during BAL treatment would significantly advance. Such information would be critical in determining treatment timing and the potential need for device replacement; both are important considerations due to the instability of hepatocellular function in many settings and the demonstrated deleterious effect of plasma from patients with liver failure on cultured hepatocytes. Even if clinical studies of the current BAL devices do not demonstrate efficacy, the information gleaned from these studies, along with improvements in cell delivery and functional stabilization, will ultimately form the basis for the next generation of devices.

## 4. Conclusions

A liver support system must provide the liver's metabolic, synthetic functions for the organ's end dysfunction, which can affect liver failure, pathophysiology, and human safety. The bio-artificial system combined modules containing live cells or hepatocytes can provide a synthetic function for the detoxification replacement and management functions in liver failure. New approaches in bio-artificial devices must focus on identifying available cell source (human) hepatocytes from stem cells, human hepatocytes, preserved or genetically engineered pig cells, or a combination of animal and human cells. Cell types for each therapy must be determined and produced in large numbers for large-scale clinical application, efficiently cultured both in vitro and biologically engineered cells, have long-term use to support liver function and show human safety, especially on xenozoonotic and tumorigenicity issues.

The regenerative treatment approach for liver disease can solve the lack of available donors. Artificial and bioartificial liver support devices have shown specific detoxifying abilities, enable original liver regeneration, and bridge the patients to transplantation, but survival effects still need further studies. It is essential to emphasize the difficulty in designing a large-scale clinical study to prove its efficacy. Further clinical application analysis for the liver support system is needed to identify the target population and suitable diseases or conditions for treatment. The effectiveness of therapy must be evaluated based on adequate and relevant clinical variable definitions (survival rate, ability to bridge liver transplantation) and not based on the potential effect at a pathophysiological end point. It is also essential to consider the effects of liver transplantation in future trial designs.

**Author Contributions:** Conceptualization, C.O.M.J.; methodology, C.O.M.J. and A.M.; validation, A.M. and R.D.A.; resources, C.O.M.J., A.M. and R.D.A.; writing—review and editing, C.O.M.J., A.M. and R.D.A.; project administration, C.O.M.J. All authors have read and agreed to the published version of the manuscript.

**Funding:** This research is funded by the Indonesia National Research and Innovation Agency Grant 2022 (Kementerian Riset dan Teknologi/Badan Riset dan Inovasi Nasional), number prn-0134341967. However, the funding body played no role in the study design and collection, analysis, data interpretation, and manuscript writing.

**Institutional Review Board Statement:** Not applicable.

**Informed Consent Statement:** Not applicable.

**Data Availability Statement:** Not applicable.

**Acknowledgments:** The authors thank Myranda Zahrah Putri for helping with the manuscript's administrative and technical preparation.

**Conflicts of Interest:** The authors declare no conflict of interest.

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
