# Peer review of "Bio-Artificial Liver Support System: A Prospective Future Therapy"

_livers, doi:10.3390/livers3010006_

Round 1

Reviewer 1 Report

In this manuscript, the authors reviewed key recent discoveries related to artificial and bio-artificial liver support systems connected with regenerative medicine. Their introduction started from artificial liver support systems to bio-artificial & iPSC-based liver support systems. For the bio-artificial & iPSC-based liver support systems, the authors also provide concise descriptions of relevant animal/human subject studies, which is followed by a well-summarized conclusion. This review thus provides useful insights for the scientific community, clinicians, and the industry.

However, the author should consider addressing the following issues in this manuscript:

Major points:

1. The authors should consider adding a table summarizing the current evidence regarding the animal- & human-subject study results for all the five types of liver support systems listed in Table 1. The table should include essential information such as the relevant animal models, materials used, parameters examined in animals, the characteristic of the patients/clinical trials, parameters examined in humans, etc. The authors should also add some descriptions in relevant sections if there are some interesting animal study results regarding the first three artificial liver support systems.

2. The authors should also consider providing a 5-panel figure, illustrating the basic components and treatment principle of these five liver support systems.

Minor points:

1. The authors should re-organize the titles of the subsections, to make the structure of the main text clearer.

2. The authors may miss some information in Fig.1. For example, according to lines 116-122, there is also a step for the screening of titles.

To summarize, this manuscript could be improved by addressing the above issues.

Author Response

Dear Reviewer, Thank you very much for the review. Here we try to answer point by point of the reviews.

Major points:

  1. The authors should consider adding a table summarizing the current evidence regarding the animal- & human-subject study results for all the five types of liver support systems listed in Table 1. The table should include essential information such as the relevant animal models, materials used, parameters examined in animals, the characteristic of the patients/clinical trials, parameters examined in humans, etc. The authors should also add some descriptions in relevant sections if there are exciting animal study results regarding the first three artificial liver support systems.

Answer: Done on lines 332-333

  1. The authors should also consider providing a 5-panel figure illustrating these five liver support systems' essential components and treatment principles.

Answer: Done added the BAL system scheme on line 267

Minor points:

  1. The authors should re-organize the titles of the subsections to clarify the structure of the main text.

Answer: Done

  1. The authors may miss some information in Fig.1. For example, according to lines 116-122, there is also a step for screening titles.

Answer: Done on lines 76-77

Reviewer 2 Report

In this review, the authors use the term "extracorporeal liver support systems (ECLS)" and consider it different from bioartificial liver support systems and artificial liver devices. Indeed, artificial devices (e.g. MARS, SPAD and Prometteus) and bioartificial liver devices are part of the family of extracorporeal liver support systems.

Line 45-46 « There is much enthusiasm for developing artificial and bioartificial support systems, 45 which have gone through many clinical trials [5] ». The authors use the reference 5 - Angeli P, Bernardi M, Villanueva C, Francoz C, Mookerjee RP, Trebicka J, et al. 2018. EASL Clinical Practice Guidelines for 379 the management of patients with decompensated cirrhosis. J Hepatol. 2018 Aug 1;69:406–60. This reference describes clinical practice in patients with decompensated cirrhosis and not the development and investigation of artificial and bioartificial liver support systems. This reference should be replaced by a more representative one.

Line 48-50- « Cell transplantation, including hepatocytes, stem cells, and liver from tissue engineering, is currently being studied.[6] ». The refernce 6 used here describes the use of induced human functional hepatocytes in the bioartificial liver devices used to treate porcine acute liver failure and not the cell transplantation ! again the reference used here should be replaced by a more representative one.

Line 54, the reference 8 describe the isolation and culture methodes of primary hepatocytes and not the systems to support liver funstions. Wrong reference..

In the results, the authors started their description with the SPAD system and compared it to the MARS system before defending and introducing it. I am not sure it is a good idea to start with SPAD, which was first studied and developed in 1999, while MARS was first developed in 1993. It does not make sense to list the systems in the wrong order of development.

The reference 11 used in line 141 describes another type of support device and not the SPAD. This is again a false reference.

In paragraph 3.1.2. - there is a great deal of confusion between the general term extracorporeal liver support systems and the term artificial or bioartificial. This paragraph is not necessary in the results and it will be better placed in the introduction provided that a clear distinction is made between artificial and bioartificial as two sub-families of extracorporeal liver support systems.

Authors give a more importance to artificial systems by going as far as to make the history and even compare the systems between them. Coming to paragraph 3.2, which is the focus of this review, they have not given value to bioartificial systems and they have passed briefly without taking into consideration the different BALs that have been developed and that have led to some clinical studies. This makes the initial focus of this review lost.

The authors have discussed iPSCs in paragraph 3.3 without having made the connection between current hepatocyte sources and the importance of iPSCs and their use. there is a lack of data on iPSCs, on their logistical importance as a potential source of hepatocytes for BAL.... they have put a section on hepatocytes differentiated from iPSCs in the animal-subject study paragraph, its placement in this section is not logical.

Author Response

Dear Reviewer, thank you very much for the constructive reviews. Here we try to answer point by point of the reviews.

  1. In this review, the authors use the term "extracorporeal liver support systems (ECLS)" and consider it different from bioartificial liver support systems and artificial liver devices. Indeed, artificial devices (e.g., MARS, SPAD, and Prometteus) and bioartificial liver devices are part of the family of extracorporeal liver support systems.

Answer: The ECLS section is deleted since it becomes the main topic of MARS, SPAD, and BAL

  1. Line 45-46 :

There is much enthusiasm for developing artificial and bioartificial support systems, 45 which have gone through many clinical trials [5] ». The authors use reference 5 - Angeli P, Bernardi M, Villanueva C, Francoz C, Mookerjee RP, Trebicka J, et al. 2018. EASL Clinical Practice Guidelines for 379 the management of patients with decompensated cirrhosis. J Hepatol. 2018 Aug 1;69:406–60. This reference describes the clinical practice in patients with decompensated cirrhosis, not the development and investigation of artificial and bioartificial liver support systems. A more representative one should replace this reference.

Answer: Done changed the reference on Lines 45-46

  1. Line 48-50: Cell transplantation, including hepatocytes, stem cells, and liver from tissue engineering, is currently being studied.[6] ». Reference 6, used here, describes the use of induced human functional hepatocytes in the bioartificial liver devices used to treat porcine acute liver failure and not cell transplantation! Again the reference used here should be replaced by a more representative one.

Answer: Done changed the reference on Line 50

  1. Line 54, reference 8 describes the isolation and culture methods of primary hepatocytes and not the systems to support liver functions—wrong reference.

Answer: Done changed the reference on Line 76

  1. In the results, the authors described the SPAD system and compared it to the MARS system before defending and introducing it. I am not sure it is a good idea to start with SPAD, first studied and developed in 1999, while MARS was first developed in 1993. It does not make sense to list the systems in the wrong development order.

Answer: Done. It has been moved from SPAD to the MARS system section first on lines 138 and 193

  1. Reference 11, used in line 141, describes another type of support device, not the SPAD. This is, again, a false reference.

Answer: Done changed the reference on Line 227

  1. In paragraph 3.1.2. - there is a great deal of confusion between the general term extracorporeal liver support systems and the term artificial or bioartificial. Therefore, this paragraph is not necessary for the results, and it will be better placed in the introduction, provided that a clear distinction is made between artificial and bioartificial as two sub-families of extracorporeal liver support systems.

Answer: Done deleted on 3.1.2 and moved the ECLS explanation to the Introduction part (lines 50-76)

  1. Authors give more importance to artificial systems by going as far as to make history and even compare the systems between them. Unfortunately, coming paragraph 3.2, which is the focus of this review, they have not given value to bioartificial systems, and they have passed briefly without considering the different BALs that have been developed and that have led to some clinical studies. This makes the initial focus of this review lost.

Answer: The given value of bioartificial systems compared to other systems have explained in Line 229 and Table 1 in Line 350.

  1. The authors have discussed iPSCs in paragraph 3.3 without making the connection between current hepatocyte sources and the importance of iPSCs and their use. There is a lack of data on iPSCs, on their logistical importance as a potential source of hepatocytes for BAL.... they have put a section on hepatocytes differentiated from iPSCs in the animal-subject study paragraph, and its placement in this section is not logical.

Answer: Done. It is clearly stated that the iPSCs can be the alternative sources of hepatocyte production for large-scale clinical trials of Bio-Artificial Liver (on the line 286-288,305-308).

Reviewer 3 Report

The authors have collected information on a very interesting and equally important research area. For the reader's sake I have a few concerns: 

1. Do include some visual illustrations in the report to make it more explanatory.

2. Make the introduction more comprehensive by adding some specific examples from the recent literature.

3. Details under all the subheads in the Results section should provide details on respective examples from previous reports

4. Table 1 needs to be more objective rather than subjective type.

Author Response

Dear Reviewer, thank you very much for the constructive reviews. Here we try to answer point by point of the reviews.

  1. Do include some visual illustrations in the report to make it more explanatory.

Answer: Done added the BAL system scheme on line 267

  1. Make the introduction more comprehensive by adding specific examples from the recent literature.

Answer: Done on lines 50-60

  1. Details under all the subheads in the Results section should provide details on respective examples from previous reports.

Answer: Done. It is clearly explained the subhead methods based on previous studies.

  1. Table 1 needs to be more objective rather than subjective type.

Answer: Done revised to be more objective by adding the number of indicator measures.

Round 2

Reviewer 1 Report

The authors have made reasonable responses to the majority of previously mentioned issues, thus the manuscript improved considerably and is now worthy of publication in Livers.